# Megapixel camera arrays enable high-resolution animal tracking in multiwell plates

Ida L. Barlow[1,2,7], Luigi Feriani[1,2,7], Eleni Minga[1,2], Adam McDermott-Rouse[1,2], Thomas James O'Brien [1,2], Ziwei Liu[1,2,5], Maximilian Hofbauer [3], John R. Stowers[3], Erik C. Andersen [4], Siyu Serena Ding[1,2,6] & André E. X. Brown [1,2✉]

Tracking small laboratory animals such as flies, fish, and worms is used for phenotyping in neuroscience, genetics, disease modelling, and drug discovery. An imaging system with sufficient throughput and spatiotemporal resolution would be capable of imaging a large number of animals, estimating their pose, and quantifying detailed behavioural differences at a scale where hundreds of treatments could be tested simultaneously. Here we report an array of six 12-megapixel cameras that record all the wells of a 96-well plate with sufficient resolution to estimate the pose of *C. elegans* worms and to extract high-dimensional phenotypic fingerprints. We use the system to study behavioural variability across wild isolates, the sensitisation of worms to repeated blue light stimulation, the phenotypes of worm disease models, and worms' behavioural responses to drug treatment. Because the system is compatible with standard multiwell plates, it makes computational ethological approaches accessible in existing high-throughput pipelines.

[1] Institute of Clinical Sciences, Imperial College London, London, UK. [2] MRC London Institute of Medical Sciences, London, UK. [3] Loopbio GmbH, Vienna, Austria. [4] Department of Molecular Biosciences, Northwestern University, Evanston, IL, USA. [5] Present address: Department of Medical Biochemistry and Biophysics, Karolinska Institute, Stockholm, Sweden. [6] Present address: Max Planck Institute of Animal Behavior, Konstanz, Germany. [7] These authors contributed equally: Ida L. Barlow, Luigi Feriani. ✉email: andre.brown@imperial.ac.uk

Recording and quantifying animal behaviour is a core method in neuroscience, behavioural genetics, disease modelling, and psychiatric drug discovery. Both the scale of behaviour experiments and the information that can be extracted from them have increased dramatically[1–4]. However, further increases in throughput are possible and would enable entirely new kinds of experiments. We therefore sought to build a system to image freely behaving animals that would maximise both phenotypic content and experimental throughput. In terms of phenotypic content, a key parameter is the resolution of the recording. If there is sufficient spatial and temporal resolution, then body parts can be identified and tracked, the animal's pose can be estimated, and the full suite of computational ethology methods can be applied to analyse any behaviour of interest. Because of its simple morphology, detailed pose estimation is well-established for the roundworm *C. elegans*[5–18] and previous work has shown the usefulness of detailed behavioural phenotyping in several domains including, for example, classifying mutants[5,6,19,20], studying chemotaxis[21] and thermotaxis[22], quantifying escape responses[23–25], and addressing basic questions in computational ethology and the physics of behaviour[10,26,27]. Maintaining sufficient resolution for pose estimation was therefore the first design constraint we required. In early worm trackers, maintaining high resolution required a motorised stage to keep a single worm in the field of view of a low-resolution camera[5,28,29], but the availability of inexpensive megapixel cameras enabled multiworm tracking with sufficient resolution to estimate each worms' pose and determine its head position[12,17]. High spatial resolution and throughput has been demonstrated using flatbed scanners to quantify worm lifespans[30] and behavioural decline with age[31]. However, the low temporal resolution (1 frame per 15 min) precludes the detection of behavioural phenotypes which happen at shorter time scales.

To maximise experimental throughput, we wanted to use off-the-shelf multiwell plates so that any behaviour screening pipeline would still be compatible with existing pipeline elements, such as liquid and plate-handling robots as well as small animal sorting machines. Because behaviour occurs over time, a standard plate-scanning approach in which each well of a multiwell plate is imaged in turn using a motorised stage limits throughput regardless of scan speed since each well must be recorded long enough to observe the behaviour of interest. Moreover, mechanical arrangements with moving parts introduce higher maintenance costs and have a higher risk of failure compared to a static camera system. Therefore, our second design constraint was that the system should be able to image all of the wells of a multiwell plate simultaneously without move parts.

Our solution to simultaneously image a large area with high resolution was to use an array of machine vision cameras that are small enough to be arranged in close proximity to one another with partially overlapping fields of view at a resolution sufficient to track small animal pose. Here we present the design of an array of six 12-megapixel cameras that uses a near-infrared light panel for illumination, a set of high-intensity blue LEDs for photostimulation, and the associated open-source software for automatically identifying wells and keeping track of metadata. The software is fully integrated into our existing Tierpsy Tracker software[17], including a graphical user interface for reviewing tracking data, joining trajectories, and annotating problematic wells to discard from analysis, as well as a neural network for distinguishing worms from nonworm objects. We demonstrate the potential of the new tracking system in neuroscience, disease modelling, genetics, and phenotypic drug screening.

## Results

**Megapixel camera array design**. Based on our previous work with single-worm tracking[13], we set a target resolution of at least 75 pixels/mm and a recording rate of 25 frames per second in order to accurately estimate worm pose and identify the head from the tail which requires the measurement of the small head swinging behaviour that is often referred to as 'foraging' in the worm community. These constraints require a total of $8100 \times 5400$ pixels, or about 44 megapixels with a 3:2 aspect ratio, to cover a standard 96-well plate. Single cameras with this resolution that can record at 25 frames per second are not available commercially. We therefore considered arrays of cameras and found that six Basler acA4024 cameras (Basler AG, Ahrensburg, Germany) in a $3 \times 2$ array equipped with Fujinon HF3520-12M lenses (Fujifilm Holding Corporation, Tokyo, Japan) was an optimal solution: This combination of lenses and cameras allowed mounting the cameras in close proximity, whereas cameras with higher resolution or larger sensors would have required significantly larger lenses. Imaging a multiwell plate with multiple cameras significantly reduces the blind spots caused by vertical separators between wells, compared with using a single camera with a conventional lens. A similar effect could be obtained by using a single camera with a telecentric lens, but the multi-camera approach remains a more compact and cost-effective way of achieving the required resolution. To provide uniform illumination whilst mitigating light-avoidance response, we used a dedicated light system using 850 nm LEDs (Loopbio GmbH, Vienna, Austria, see Material and Methods for more details). Blue light stimulation is provided by a custom LED array (Marine Breeding Systems, St. Gallen, Switzerland) using four Luminus CBT-90 TE light-emitting diodes (bin J101, 456 nm peak wavelength, 10.3 W peak radiometric flux each) with user-tunable intensity. The camera lenses are equipped with long pass filters (Schneider-Kreuznach IF 092, Schneider-Kreuznach, Germany, and Midopt LP610, Midwest Optical Systems Inc, Palatine, IL, USA) to block the photostimulation light while allowing the brightfield IR signal to reach the camera sensors. A schematic of the imaging system is shown in Fig. 1a–c. To further increase throughput, we built five units of the camera arrays which can be operated in parallel (Kastl-High-Res, Loopbio GmbH, Vienna, Austria).

**Choice of suitable multiwell plate design**. Because the fields of view of the six cameras partially overlap, the imaging system provides flexibility in selecting a multiwell plate with any number of wells. For our purposes, 96-well plates with square wells provided a good balance between imaging area and the number of wells (Fig. 1d, e). Plates with smaller numbers of wells would reduce imaging throughput, while 96-well plates with circular wells would reduce the area available for worms to behave and increase shadowing around the well edges (Supplementary Fig. 1a, b). Using square well plates (Whatman 96 well plate with flat bottom, GE Healthcare, Chicago, IL, USA) significantly increases the efficiency of the system: in our tests, in plates with circular wells only 21% of the imaging area is available for capturing useful data, while the rest is outside of any well or lost in shadows. For square wells, 43% is available for behaviour. This has important implication for tracking and experimental design, as not all worms placed in a well will be always visible. For example, when imaging worms of the N2 control strain, all worms in a well are simultaneously tracked 40% of the time, and this figure can depend on the strain, with worms of the wild isolate strain CB4856 being all visible simultaneously only 9% of the time (Supplementary Fig. 5). The fraction of the imaging area available for tracking can be further increased by using custom plates with thinner wall dividers and shallower wells to reduce the

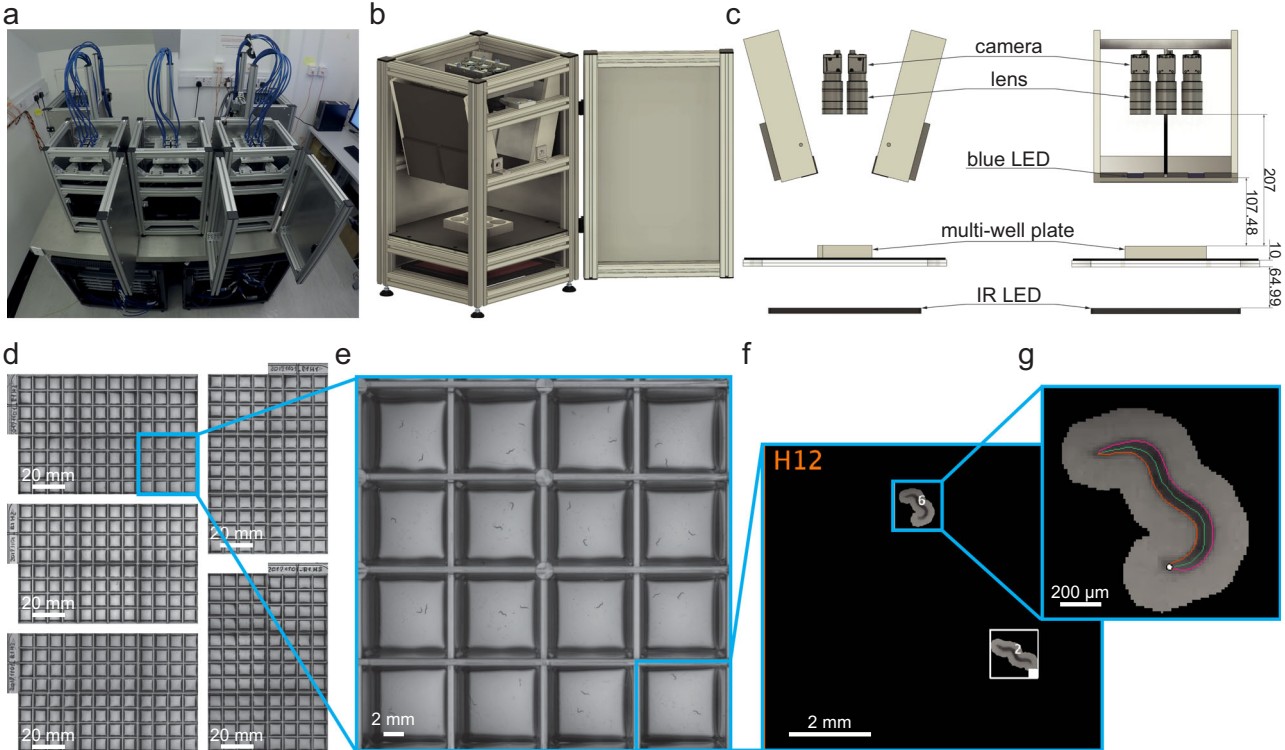

**Fig. 1 Schematic of megapixel camera arrays. a** Five identical camera arrays were mounted on an air supported table. The associated workstations to run the Motif software were arranged in the two server racks underneath. **b** 3D schematic drawing of a single imaging unit (Kastl–Highres). The six cameras were mounted on a plate that is connected to the rig frame by three spring-loaded screws, and can be moved along the vertical axis. This allows for changing the focal plane of all six cameras at once. One of the imaging unit's side panels is omitted from this view. **c** Technical drawing (front and side view) of an imaging unit annotated with dimensions in millimetres. **d** Using five identical units, 480 wells can be recorded simultaneously. Zooming in to the (**e**) camera, (**f**) well, (**g**) and worm level shows that this system achieves enough resolution to precisely track the nematodes. Each square well measures 8 × 8 mm.

shadowing (Supplementary Fig. 1c), but this comes at an increased cost of manufacture.

The output of the combined system is 30 videos tiling across the five imaged multiwell plates corresponding to 480 simultaneous behaviour assays (Fig. 1d). Expanding the image to the level of a single well (Fig. 1f) and a single animal (Fig. 1g) shows that the resolution is sufficient to estimate the pose and identify the head of single worms, which can reveal detailed trajectory differences between individuals that are the basis for quantitative behavioural phenotyping.

**High throughput imaging.** Due to the high amount of raw image data produced by USB3 cameras at full bandwidth (6 cameras recording at 25 fps produce approximately 6.5TB/hour of raw footage), live compression on a dedicated system was required. To achieve this, we used a total of 10 Motif Recording Units (Motif— Video Recording System, Loopbio GmbH, Vienna, Austria) equipped with Nvidia Quadro P2000 GPUs (Nvidia Corporation, Santa Clara, CA, USA), each recording from 3 cameras. The two recording units with cameras from the same system were set up in a parent-child configuration.

The Motif software acquires and compresses images on the fly and stores them in the open imgstore format (https://github.com/loopbio/imgstore) along with timestamps and frame numbers for each individual frame, as well as continuous and synchronised recordings of environmental data (for each unit this was: outside temperature and inside temperature, humidity, and light level). Recording the time and frame number for each image allows precise timekeeping over a long recording duration as it removes temporal drift due to skipped or dropped frames and due to

differences in camera clocks. Additional metadata for each recording is saved with the imgstore, including the camera serial numbers, camera and system settings, and any user-defined data.

A single workstation manages all imaging experiments on all units across the whole system, from experimental parameter tuning (including the intensity of the photostimuli) to video collection to data transfer, by accessing the Motif user interfaces using the web browser of the parent machines. Given the large number of high-resolution cameras, the control workstation was connected to a large monitor (we use a 43-inch 4k monitor) to facilitate camera focussing and sample positioning.

In addition to providing a web-accessible user interface, Motif allows complete control of the camera arrays and arbitrarily complex scheduling of data acquisition and photostimulation programmatically, via an API (https://github.com/loopbio/python-motifapi). This allows us to run imaging experiments on all camera arrays by executing a single Python script on the monitoring workstation. Encoding the parameters of experiments in a script improves reproducibility by making settings consistent over time by default.

**Updates to Tierpsy Tracker, and companion software, for multiwell imaging format.** In our camera array system, each camera records multiple wells which complicates metadata handling since there is no longer a one-to-one correspondence between a video file and a particular experimental condition. We have updated Tierpsy Tracker[17] to handle videos with multiple wells: it can automatically identify wells from the video (Fig. 1f), and return results on a well-by-well basis. In the Viewer, the user can see the names and boundaries of the wells, and have the

option of marking any well as "bad" if necessary. This flag is propagated to the final tracking results so that the contribution of "bad" wells can be filtered out for downstream analysis.

To keep track of the experimental conditions of each well we have developed an open-source module in Python to automatically handle experimental metadata (github.com/Tierpsy/tierpsy-tools-python). For each experiment, a series of csv files specifies the worms and compounds (if applicable) that were added to each well. This can include information on how a COPAS worm sorter (Union Biometrica) was used to dispense different strains in the wells of the imaging plates, which compound source plate was replicated onto each imaging plate, or any column shuffling performed by a liquid handling robot. These tables are then combined to create a mapping between each well in an imaging plate (identified by a unique ID) and an experimental condition. For each imaging run, the user needs to log the camera array used for each imaging plate at the time of the experiment. This information is then mapped to video file names to create a final metadata table suitable for subsequent analysis (see Material and Methods and Supplementary Fig. 3 for more details).

Another key software improvement we incorporated is a convolutional neural network (CNN) to exclude nonworm objects from subsequent analysis. While we previously used contrast-based segmentation and size-based filtering for worm detection in our analysis[17], introducing the CNN into Tierpsy Tracker improves the quality of the tracking data and the subsequent analysis results as well as the speed of the analysis because fewer objects are analysed in subsequent steps (see Methods for more details).

Tierpsy Tracker does not maintain the identity of the worms across gaps in tracking which can occur when worms leave the field of view or cross each other. The number of unique tracks is thus typically greater than the number of worms. While Tierpsy initially calculates features on a track-by-track basis, we use a single averaged feature vector per well because of the uncertainty in how the tracks map to individual worms. The natural unit of measurement of sample size with the setup described in this work is therefore a well, rather than an individual worm.

**Rapid assessment of natural variation in behaviour**. We tracked the behaviour of N2 and wild isolates of the divergent set in the *C. elegans* Natural Diversity Resource (CeNDR) strain collection with our system to detect natural variation in behaviour[32]. To further increase the dimensionality of the behavioral phenotypes, we included a blue light stimulation protocol using a set of four bright blue LEDs. Each tracking experiment is divided into three parts: 1) a 5-minute pre-stimulus recording, 2) a 6-minute stimulus recording with three 10-second blue light pulses starting at 60, 160, and 260 sec, and 3) a 5-minute post-stimulus recording. Blue light can elicit an escape response in worms, thus expanding the range of observable behaviours[33,34]. Programmable blue light stimulation is reproducible, compatible with high throughput assays, and is also useful for optogenetic stimulation.

We tracked on average 20 wells per strain. Given the high throughput achieved with our new system, this experiment can be performed within a few hours. The recordings of the camera array maintain enough resolution to extract the full set of Tierpsy features[35], which describe in detail the morphology, movement, and posture of the worms, including subdivision by motion mode (forward, backward, and paused) and body part. We extract a set of 3076 summary features per well for each recording period (pre-stimulus, blue light, and post-stimulus), resulting in a total of 9228 features for each well. This allows us to detect fine differences in the morphology, posture and movement of the worms which varies in a nonuniform way among wild isolates

(Fig. 2a-c). The neck curvature of wild isolates tends to show more significant differences to N2 worms than the curvature of other parts of the body, which might be related to differences in foraging behaviour between N2 and wild isolates (Fig. 2b). However, not all strains show the same curvature pattern across the body indicating natural variation in posture. All the wild isolates move on average faster than N2 worms but their response to blue light varies (with some being more and others less sensitive to blue light), showing that the blue light stimulus increases the dimensionality of behavioural differences (Fig. 2c).

To assess how well we can predict the worm strain based on its behavioural fingerprint, we estimated the classification accuracy using a random forest classifier. We first split the data into a training/tuning set and a held-out test set. We used the training set to select features using recursive feature elimination (RFE) and tune the hyperparameters of the model. Figure 2d shows the highest cross-validation accuracy achieved for different sizes of selected feature sets. The accuracy improves when we select features increasing the samples-to-features ratio, as this helps control overfitting and, in parallel, reduces the correlation between features. Combining features from different blue light conditions (blue curve) increases the dimensionality of the data and the classification accuracy. Using the best performing features and hyperparameters, we trained a classifier with the entire training/tuning set and used it to make predictions in the test set. The test accuracy we achieved is 66% which is significantly higher than random (9%).

**Temporal response and sensitisation to aversive blue light stimulation**. Having established that blue light stimulation can be leveraged to improve classification accuracy, we moved to further investigated the response elicited by blue light in N2 and CB4856 at a higher temporal resolution.

We imaged with blue light stimulation on both the N2 and CB4856 strains, and observed different behavioural responses between these two strains. We extracted the same set of 3076 previously defined features[35] with a time resolution of 10 sec and used these to construct the behaviour phenotype space. N2 and CB4856 have well-known behavioural differences[26,36–40] and are expected to occupy different regions of the phenotype space. Principal component analysis (PCA) shows that application of blue light stimulation moves the strains from their already distinct positions in the plane defined by the first two principal components (PCs) to new positions, indicating detectable responses to the stimulus in both strains (Fig. 3a). Blue light-induced displacement through phenotype space led to better separation between the two strains (Fig. 3a, right), confirming that the addition of the stimulus can reveal further behavioural differences between two strains already known to be distinct.

The *C. elegans* escape response is characterised by a combination of increased forward locomotion and decreased spontaneous reversals[41]. The differences in blue light-induced escape response between the N2 and CB4856 strains can thus also be seen by simply examining the fraction of the worm population moving forwards, moving backwards, or remaining stationary. For both strains, a single photostimulus triggers a sharp and steady increase in the fraction of worms moving forwards, followed by a relaxation towards the pre-stimulus level once the stimulus ceases. The fraction of worms moving backwards has a slight increase at the beginning of the stimulus, and then decreases without increasing again until the stimulus is over. Finally, the fraction of stationary worms declines rapidly during the stimulus and is restored after the stimulus ends (Fig. 3b). However, while in CB4856 the rate at which the population fractions return to the pre-stimulus levels is steady, N2 shows a sharp initial decline in

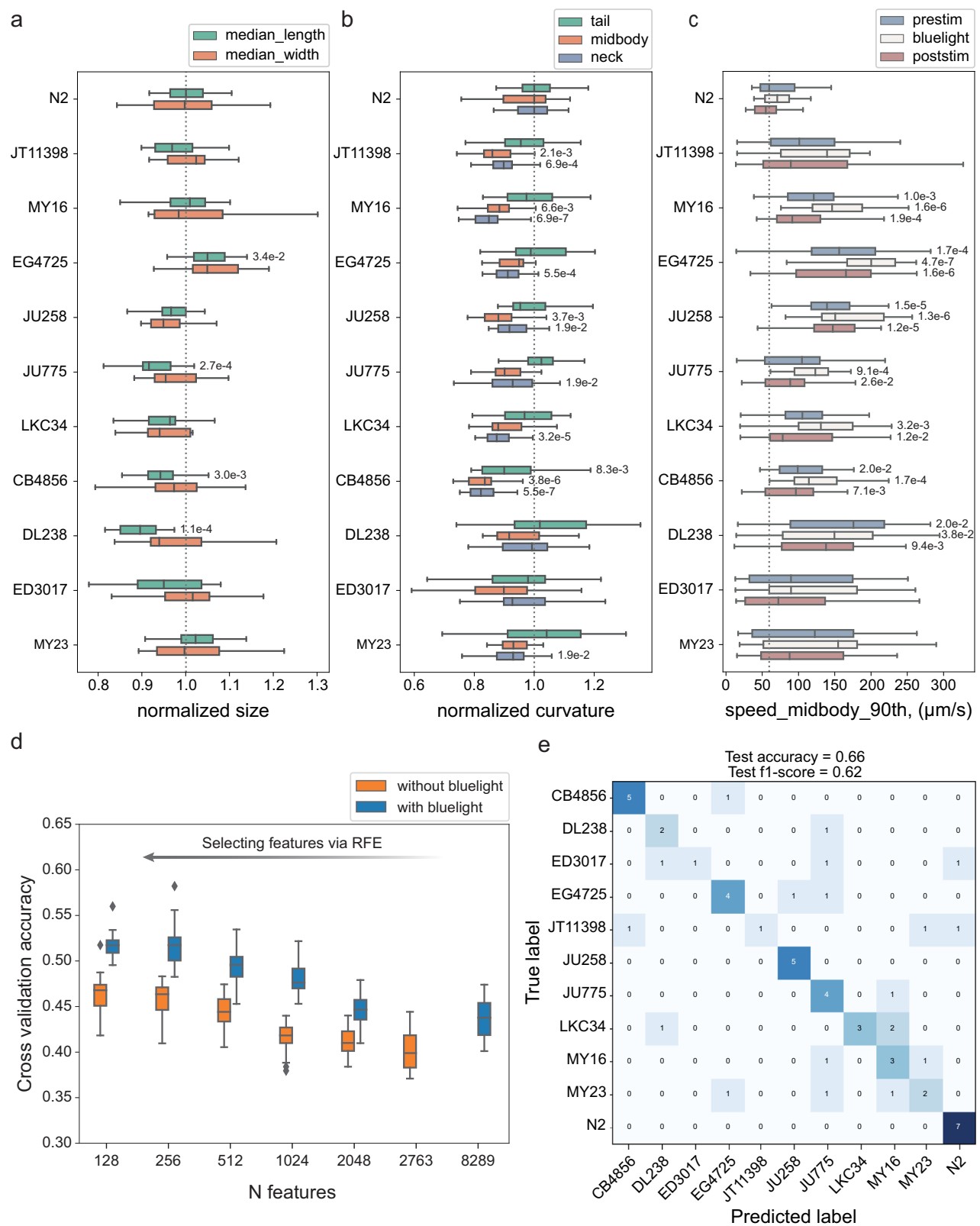

forward-moving worms over several seconds (and a corresponding sharp increase in stationary worms and backward-moving worms) before relaxing steadily. Repeated photostimulation (twenty pulses of 10 s on, 90 s off) of N2 worms causes sensitisation, as light pulses trigger a progressively increasing fraction of worms to move forwards (Fig. 3). Meanwhile, between light pulses, worms recover to a progressively decreasing baseline level of forward locomotion (and conversely, progressively increasing stationary fraction), possibly due to fatigue from increased activities during the pulse. This reduced forward locomotion fraction persists in the absence of photostimulation, with no obvious return towards pre-stimulus levels over a 6.5 min period after the final pulse (Fig. 3c, e). The combined effect of sensitisation and fatigue leads to a roughly linearly increasing

**Fig. 2 Natural variation in behaviour. a–c** Examples of features describing morphology, movement, and posture. Each box shows median, 25th and 75th percentile (central mark, left and right edge, respectively), while whiskers show the rest of the distribution except for outliers (outside 1.5 times the IQR above the 75th and below the 25th percentile). Numbers near the boxplots are the p-values indicating statistically significant differences between N2 and wild isolates at a false discovery rate of 5% using Kruskal-Wallis tests and correcting for multiple comparisons with the Benjamin-Yekutieli method. P-values are omitted for nonsignificant differences. **a** Morphological differences were detected between strains. The length and the midbody width varied in a nonuniform way among strains. **b** Adequate resolution enabled detailed characterisation of the worm posture and the detection of differences among strains in multiple dimensions. The curvature at different parts of the body varied in a nonuniform way among strains. The neck curvature showed more significant differences. The parts of the body are defined following the conventions adopted in Tierpsy Tracker[35]. **c** The speed of wild isolates was on average higher than the speed of N2 worms. The response of wild isolates to blue light stimulus varied; some strains (e.g. EG4725) were more sensitive to blue light compared to N2, while others showed less obvious escape response (e.g. DL238). This provided additional dimensions to the behavioural phenotype. **d** Using the quantitative behavioural phenotypes, strains were classified with significantly higher accuracy than random. Combining features from different blue light conditions increased the dimensionality of the data and the classification accuracy between strains. **e** Worm strains were predicted in a held-out test set with 66% accuracy which is higher than random (9%). Sample size, in wells (3 worms per well): $N_{N2} = 34$, $N_{JT11398} = 21$, $N_{MY16} = 27$, $N_{EG4725} = 29$, $N_{JU258} = 25$, $N_{JU775} = 25$, $N_{LKC34} = 27$, $N_{CB4856} = 29$, $N_{DL238} = 16$, $N_{ED3017} = 20$, $N_{MY23} = 23$.

response over multiple light pulses, as illustrated by taking the difference between the fraction of worms moving forward before and after stimulation (Fig. 3d).

Photostimulation can thus better distinguish between worm strains using existing predefined feature sets, as well as create new features for quantifying the details of the escape behaviour. Similar experiments on habituation to repeated mechanical stimulation have been used extensively to study learning in *C. elegans*[42–44]. Aversive blue light stimulation acts through different sensory neurons and converges on the same motor circuits and so may provide useful comparative data to investigate the genetics and neuroscience of learning mechanisms.

The addition of these new and interpretable features increases the dimensionality of the worm behavioural phenotypic space, which may be useful for phenotyping applications.

**Behavioural phenotypes of ALS disease models in response to blue light.** A previous study generated several Amyotrophic Lateral Sclerosis (ALS) disease model strains that carry patient amino acid changes in the *C. elegans sod-1* gene[45]. This study found that the disease model strains have no obvious behavioural defects unless they are exposed to oxidative stress by overnight treatment with paraquat.

We phenotyped these ALS disease model strains on our system and saw similar results. PCA of a pre-defined set of 256 Tierpsy features[35] under standard imaging conditions (5 min of spontaneous behaviour) does not show clear differences between the strains (Fig. 4a). Adding blue light pulses (three 10-second blue light pulses over six minutes) leads to better separation between the strains in PC space (Fig. 4b). Although the SOD-1(+) wild-type control strain (blue) and the SOD-1(A4V) mutant disease strain (orange) clearly separate into their own clusters, SOD-1(H71Y), SOD-1(G85R) and SOD-1(0) null strains cluster together, suggesting that their overall responses to blue light are similar to each other. The clustering of SOD-1(H71Y), SOD-1(G85R), and SOD-1(0) strains upon blue light stimulation is consistent with the previous finding that all three strains have loss of *sod-1* function in glutamatergic neurons. By contrast, the SOD-1(A4V) strain has overexpression of *sod-1* in cholinergic neurons without affecting glutamatergic neurons[45], and this disease strain forms its own cluster in the blue light PC space (Fig. 4b).

In the previous study and in our results, no difference is observed in the baseline behaviour of the strains. However, exposure to aversive conditions, possibly acting through very different mechanisms, highlights the difference between strains by exposing an otherwise cryptic phenotype. Upon blue light stimulation, SOD-1(H71Y), SOD-1(G85R), and SOD-1(0) strains show significantly bigger increases in forward locomotion compared to the SOD-1(+) control strain and the SOD-1(A4V)

disease strain (Fig. 4c,d). This increase in forward movement appears to be primarily at the expense of stationary (Fig. 4e) rather than backwards locomotion (Supplementary Fig. 2). Nevertheless, a closer look at reversal frequencies at a finer temporal resolution reveals decreased reversals in the three *sod-1* loss-of-function strains but not the other two strains (Supplementary Fig. 2b).

**Phenotypic screen of human-approved drugs.** We used a library of 245 drugs to quantify worms' responses to human-approved drugs across multiple behavioural features. 240 drugs were from the Prestwick *C. elegans* library, a collection of small-molecule out-of-patent drugs selected by the supplier (Prestwick Chemical, Illkirch-Graffenstaden, France) for their chemical structural diversity and good tolerability in worms. To these, we added a set of 4 antipsychotics and an insecticide that have a strong phenotype that we could use as positive controls and for checking the automated metadata handling code[46]. Three worms were added to each well of 96-well plates and were left on the drug for four hours before imaging. We processed the videos using Tierpsy Tracker, extracting 3016 behavioural features[35] from each imaging condition (pre-stimulus, blue light stimulus, post-stimulus) and concatenated the feature vectors so that each well was represented by a 9048-dimensional feature vector. We used a linear mixed model to identify compounds that had a significant effect on behaviour in at least one feature[46]. The linear mixed model used the imaging day as a random effect to account for day-to-day variation in the data. The 153 compounds that had a detectable effect were kept for further analysis. We restricted the feature space to a subset of 256 features (Tierpsy256) that we selected in the previous work[35] for each imaging condition, so that each well was now represented by a 768-dimensional feature vector. The features were then z-normalised and both features and samples were hierarchically clustered using complete linkage and correlation as the similarity measure (Fig. 5a).

The compounds in the library are mostly well-characterised with known modes of action. By examining clusters in detail, we found several clusters that included multiple compounds from the same mode of action (Fig. 5b-d). One of the identified clusters contains several compounds that inhibit muscle contraction and/ or are related to vasodilation (Fig. 5b). The most clearly defined cluster (Fig. 5c) contains antibiotics, and most of them share the same mode of action (ribosomal protein synthesis inhibition). Because the worms were imaged on a lawn of bacterial food, the most likely cause of these behavioural differences is a change in the bacterial food lawn that worms sense and respond to, but a direct effect on the worms is not impossible since *C. elegans* do respond to some antibiotics[47,48]. A third cluster is formed by compounds used to treat the symptoms of Parkinson's Disease

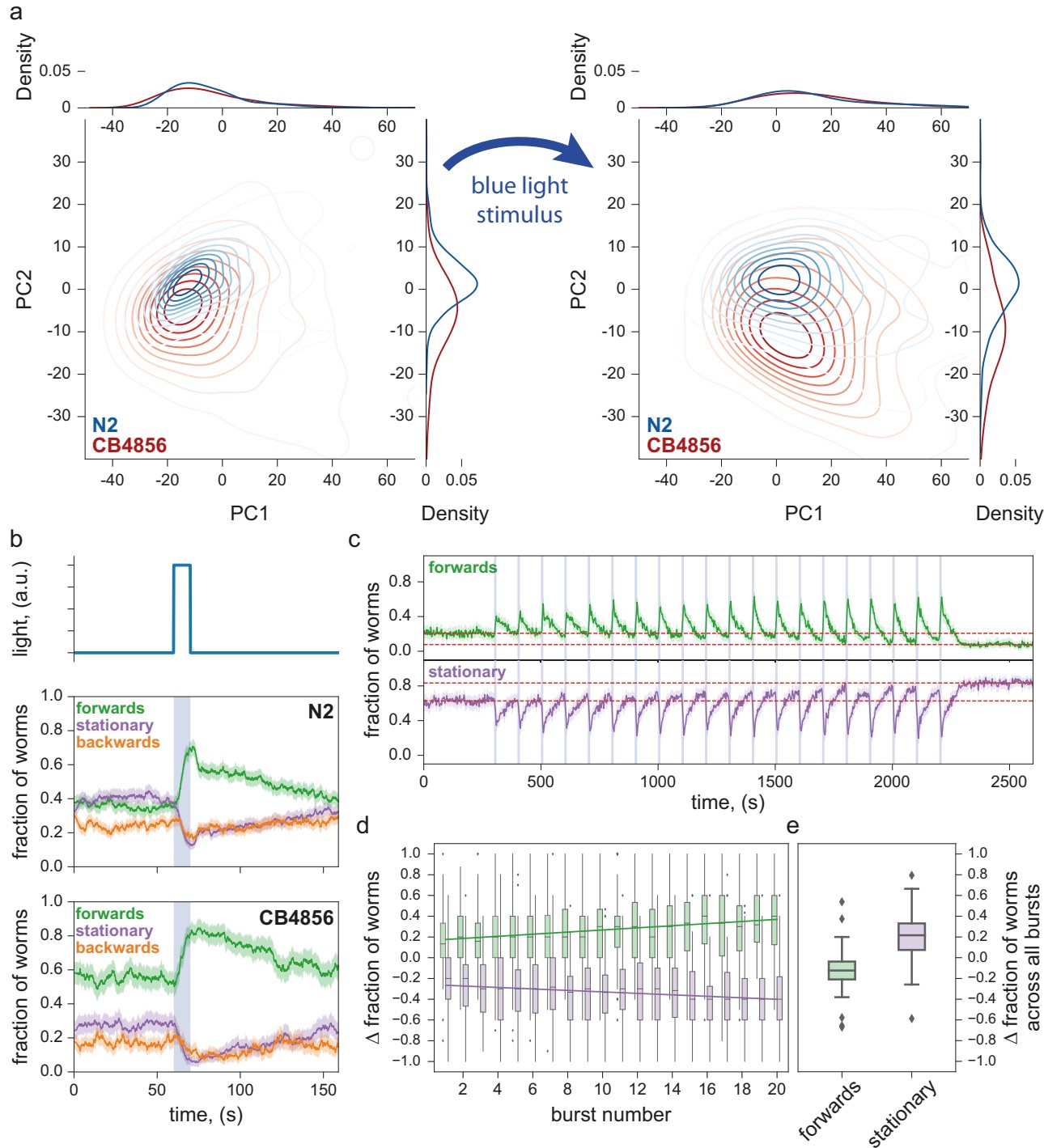

(anticholinergics). Previous studies have shown that anticholinergic drugs can affect locomotion in *C. elegans*[49,50] and also induce motor activity in other model animals[51,52].

Most of the compounds had a detectable effect on behaviour, but many of the effects were less obvious than a library of invertebrate-targeting compounds that we screened recently using the same method[46]. A part of the explanation is likely to be a lack of conservation of some drug targets between humans and worms, although it should be noted that many are sufficiently conserved that human-targeted drugs have effects through the expected receptor class[53]. Another reason some compounds do not have a detectable effect is drug uptake which is known to be an issue for drug screens in worms[54], highlighting the continuing

need for improved drug delivery to maximise the usefulness of worms in drug screening[55].

## Discussion

We have developed a megapixel camera array system to enable high throughput, high content imaging of worms in standard multiwell plates. By partially overlapping the fields of view of six cameras, we can image an entire multiwell plate at spatial and temporal resolutions that are sufficient for tracking *C. elegans* and extracting high-dimensional phenotypic fingerprints. While the experiments presented in this work were carried out in 96-well plates, the imaging system can easily support 24- and 48-well

**Fig. 3 Escape response to photostimulation. a** PCA plots of N2 and CB4856 in the 10 sec immediately before (left) and immediately after (right) a 10 s stimulus showing detectable behavioural responses: both strains moved to new, better separated positions in the phenotype space as a result of stimulation. Sample size (in wells, two worms per well) is $N_{N2} = 377$, $N_{CB4856} = 115$ (left), $N_{N2} = 398$, $N_{CB4856} = 98$ (right) **b** Photostimulation with blue light elicited similar escape responses in both N2 and CB4856 strains, with the fraction of worms moving forwards increasing during the stimulus and decreasing after the stimulus. However, post-stimulus recovery appears to occur at two timescales for N2 but not for CB4856. Solid lines are means, shaded areas show the 95% confidence interval. Sample size (in wells, two worms per well) is $N_{N2} = 529$, $N_{CB4856} = 396$. **c** Repeated photostimulation triggered increasing aversive response in N2, also leaving a higher fraction of worms stationary after serial stimulation than before (vertical separation between the two dashed lines to contrast the before and the after levels). $N = 144$ wells (three worms per well). **d** The fraction of worms triggered to move forwards by each stimulus increased throughout the stimulation series, as a decreasing fraction of worms remained stationary across each successive photostimulus. Each data point was obtained by taking the difference in a 10 s window just before and just after the end of each stimulus. $N = 144$ wells (3 worms per well). **e** After repeated photostimulation, a larger fraction of the population than before was stationary. This was quantified by taking the difference of the population fractions in each motion mode between the final 5 min and the initial 5 min of the experiment (red dashed lines in **c**). Each box shows median, 25th and 75th percentile (central mark, lower and upper edge, respectively), while whiskers show the rest of the distribution except for outliers (outside 1.5 times the IQR above the 75th and below the 25th percentile), plotted individually. $N = 144$ wells (three worms per well).

plates as well. We have added features to Tierpsy Tracker to make it compatible with the multiwell imaging format, so that each well is detected and analysed separately. We incorporated strong blue LED lights into the camera array system to provide precise and repeatable photostimulation and found that this leads to better separation between wild isolates and ALS disease model strains, in the latter case revealing phenotypes that could not be detected in standard unstimulated assays. Repeated blue light stimulation also revealed a novel sensitisation phenotype in N2 worms, in marked contrast to the habituation in the reversal response reported in previous experiments on repeated mechanical stimuli which are used to study learning in worms[12,43].

Our imaging hardware and analysis software are designed to support high throughput phenotypic screening, as the multiwell format allows for a large number of experiments to be conducted simultaneously. Furthermore, our experimental pipeline uses liquid handling robots for dispensing agar, food, drugs, and worms, in order to streamline the workflow for large-scale phenotypic screening. On a typical eight-hour imaging day, a single experimenter can operate five runs on all five-camera array units, thus collecting imaging data from 2400 independent wells in a 96-well plate format. Typical post-acquisition processing time for this volume of data (assuming the standard 16 min video length at 25 fps, three worms per well) is 50-85 h using a MacPro (Processor: 2.7 GHz 12-Core Intel Xeon E5; Memory: 64GB 1866 MHz DDR3) or 5–11 h on a local cluster to go from raw video data to fully extracted behavioural features. Processing time increases significantly with object number and depends on the quality of the video (good contrast, lack of debris, etc.).

Multiple cameras have previously been used to record large areas for worm tracking, but with an emphasis on long time recording at lower resolution compared to the applications presented here[56]. Multicamera imaging systems have also been used to record the behaviour of other species. When imaging animals in the field, for example, experimenters have employed multiple cameras to simultaneously image locations of interest[57]. The most common scenario sees the use of multiple cameras pointed at the same animals for tracking position and pose in three dimensions[58–62]. Multiple camera systems have also been used to increase coverage[63] and throughput[64,65] in Drosophila imaging experiments.

There are several options available to record data from multiple cameras, from the video acquisition tools and SDK provided by camera manufacturers to third-party tools like The Recorder (MultiCamera.Systems LLC, Houston, TX, USA) or the open-source Micro-Manager. The introduction of the GenICam Standard, developed by the European Machine Vision Association[66], has provided developers with a common programming interface that is not strictly coupled to the interface technology of the different cameras, thus making it easier to develop user interfaces.

Setting up a user-friendly multicamera system from scratch still requires a considerable investment in terms of time and know-how. A Motif camera system provides a compact imaging setup and a computer system with matched specifications and includes source code in Python, making it more transparent than some other proprietary systems.

A main strength of our camera array system is its scalability. Screening throughput can be readily expanded with additional imaging units, as the system is modular and each camera array has a relatively small physical footprint. On-the-fly compression of raw videos provided by the software keeps the data volume manageable. Post-acquisition analysis is easily parallelised since videos can be analysed independently and processing time can be decreased linearly by allocating more computational cores to the task (e.g. by using a high-performance cluster).

The megapixel camera arrays we describe here represent a natural progression in worm tracking hardware where advances in the past have come from multiplexing to increase throughput[13] and increasing resolution to get more information from multi-worm trackers[12]. Our new system will make it possible to do higher throughput screening with a resolution that enables the full suite of computational ethology tools to be brought to bear on phenotyping. We anticipate this will open new directions in large-scale behaviour quantification with applications in genetics, disease modelling, and drug screening.

## Methods

**Worm strains.** *C. elegans* strains used in this work are listed in Supplementary Table 1. Worms are cultured on Nematode Growth Medium (NGM) agar at 20 °C and fed with *E. coli* OP50 following standard procedures[18].

**Standard phenotyping assay.** The standard phenotyping assay was used for most experiments in this work unless otherwise noted (detailed protocol: https://doi.org/10.17504/protocols.io.bsicncaw). See Supplementary Table 2 for the detailed protocols used to collect the data shown in each figure panel.

Briefly, Day 1 adult worms were obtained by bleach-synchronisation (detailed protocol: https://doi.org/10.17504/protocols.io.2bzgap6) and used for all imaging experiments. Imaging plates were prepared by filling 96 well plates with 200 μL of low peptone (0.013% Difco Bacto) NGM agar per well using an Integra VIAFILL reagent dispenser (INTEGRA Biosciences Ltd, UK) (detailed protocol: https://doi.org/10.17504/protocols.io.bmxbk7in), and stored at 4 °C until use. On the day before imaging, plates were placed in a LEEC BC2 drying cabinet (LEEC Ltd, Nottingham, UK) to lose 3–5% weight (starting weight 59 g without lid, target weight 56 g, this takes between 2 and 3 h). Each plate was then seeded with 5 μL per well of 1:10 diluted OP50 using VIAFILL, and stored at room temperature overnight.

On imaging day, synchronised Day 1 adults were washed in M9 (detailed protocol: https://doi.org/10.17504/protocols.io.bfqbjmsn) and dispensed into imaging plate wells using COPAS 500 Flow Pilot worm sorter (detailed protocol: https://doi.org/10.17504/protocols.io.bfc9jiz6). Three worms were placed into each well unless noted otherwise. Plates were returned to a 20 °C incubator for 1 h to dry following liquid handling, and then placed onto the multi-camera tracker for 0.5 h to acclimatise prior to image acquisition.

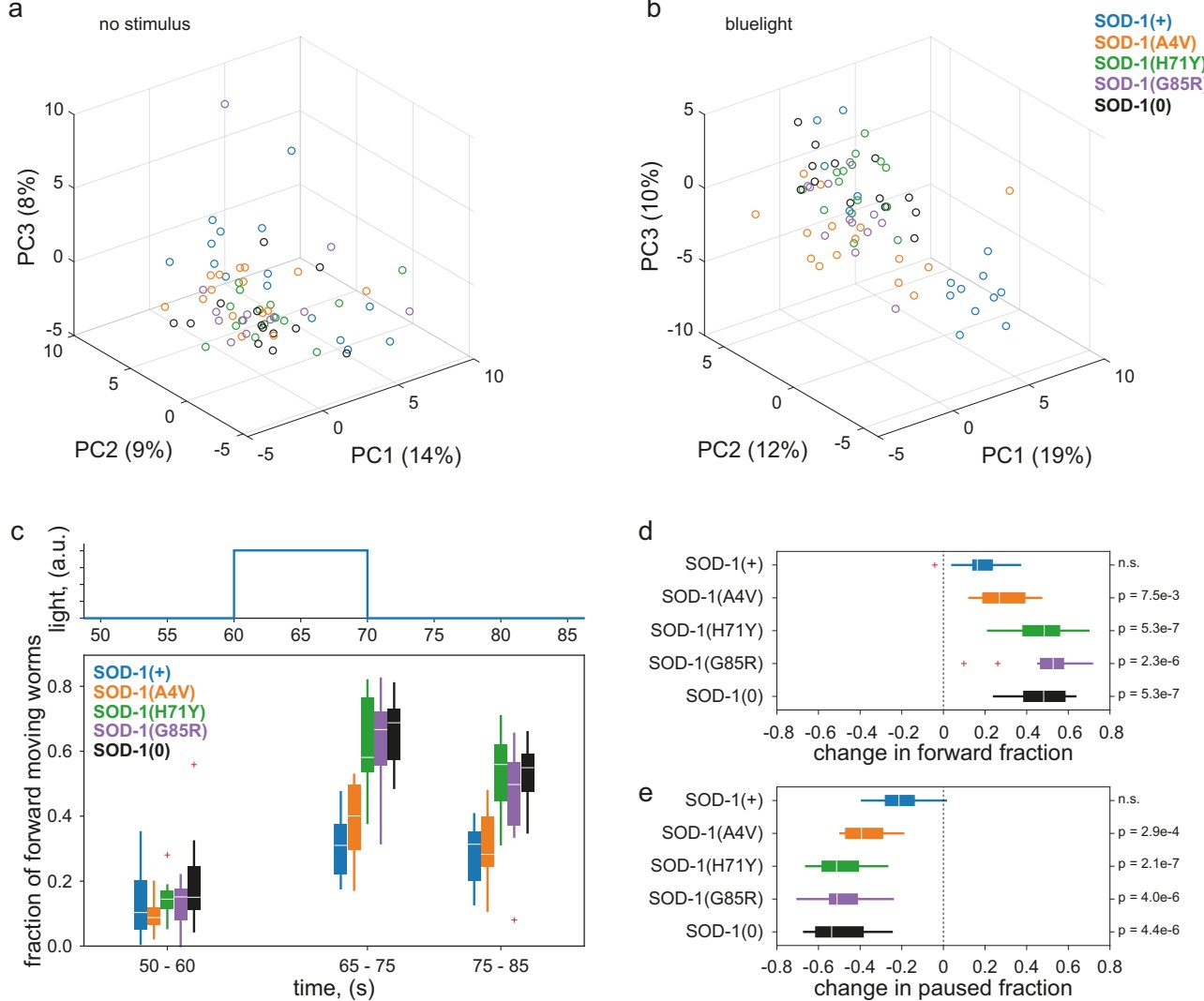

**Fig. 4 Blue light stimulation elicits different responses amongst ALS disease models. a–b** Principal component analysis of 256 extracted behavioural features from standard (**a**) or blue light (**b**) imaging conditions. Features were extracted by Tierpsy Tracker. Each datapoint represents one plate average of the strain, with up to 12 independent wells for each strain in every 96-well plate. Each well contained an average of three worms. The time window represented in B is also shown in (**c**). **c** Overall fraction of forward locomotion under blue-light imaging conditions. A 10 s blue light pulse started at $t = 60$ s and feature values were calculated using 10 s windows centred around 5 sec before, 10 s after, and 20 s after the beginning of each blue light pulse. Plate averages were used to generate the plot for each strain. Each box shows median, 25th and 75th percentile (central mark, lower and upper edge, respectively), while whiskers show the rest of the distribution except for outliers (outside 1.5 times the IQR above the 75th and below the 25th percentile), plotted individually. **d–e** Changes in the overall fraction of forward (**d**) or paused (**e**) locomotion upon blue light stimulation. The difference was calculated by subtracting the average feature values over the $t = 50$-60 sec pre-stimulus window from those over the $t = 65$-75 sec blue light pulse window (these correspond to the first and the second time points in (**c**). Plate averages were used to generate the plot for each strain. Two sample t-test compared to the SOD-1(+) control strain (n.s. not significant). Each box shows median, 25th and 75th percentile (central mark, left and right edge, respectively), while whiskers show the rest of the distribution except for outliers (outside 1.5 times the IQR above the 75th and below the 25th percentile), plotted individually. Sample size (in wells, three worms per well): $N_{sod-1(+)} = 232$, $N_{sod-1(A4V)} = 228$, $N_{sod-1(H71Y)} = 211$, $N_{sod-1(G85R)} = 165$, $N_{sod-1(0)} = 195$.

**Drug experiments**. Drug experiments followed the standard phenotyping assay workflow, but with a few modifications. A detailed protocol can be found at https://doi.org/10.17504/protocols.io.bs6znhf6.

Briefly, imaging plates were prepared with drugs the day before imaging and stored in the dark overnight at 4 °C. Using a COPAS 500 Flow Pilot, three worms were dispensed into each well of 96-well plates. Following liquid handling, plates were kept in a 20 °C incubator for an extra 3 hours to allow drug exposure (total drug exposure time was thus four hours).

**Bright field illumination**. To prevent light-avoidance response while illuminating the sample in bright field, we used a dedicated 850 nm light system (Loopbio GmbH, Vienna, Austria), consisting of a 200 × 200 mm edge-lit LED panel. Briefly, the edge-lit configuration has the LEDs attached to the frame of the panel and shining into a horizontal light guide plate, which homogenises and diffuses the light. This setup was characterised to provide a minimum radiance of 15 W m$^{-2}$ sr$^{-1}$, a minimum

uniformity of 95 ± 10%, and a half-angle (the angle at which the measured intensity falls to 50% its maximum value) is 30°.

The panel is significantly larger (200 ×200 mm) than the sample area. This achieves the double goal of minimising edge effects and thermally insulating the sample from the LED panel, as this can be placed at a relatively large distance (65 mm). To further improve light collimation, two computer monitor privacy filters (3 M, Saint Paul, MN, USA) placed at right angles are inserted between the LED panel and the sample area.

**Image acquisition**. All videos were acquired at 25 fps on the trackers in a temperature-controlled room at 20 °C, with a shutter time of 25 ms, and 12.4 μm px$^{-1}$ resolution. For all experiments unless otherwise noted, three sequential videos were taken, run in series by a script: a 5-minute pre-stimulus video, a 6-minute blue light recording with 10-second blue light pulses set to 100% intensity at the 60, 160, and 260 s mark, and a 5-minute post-stimulus recording. The timing of

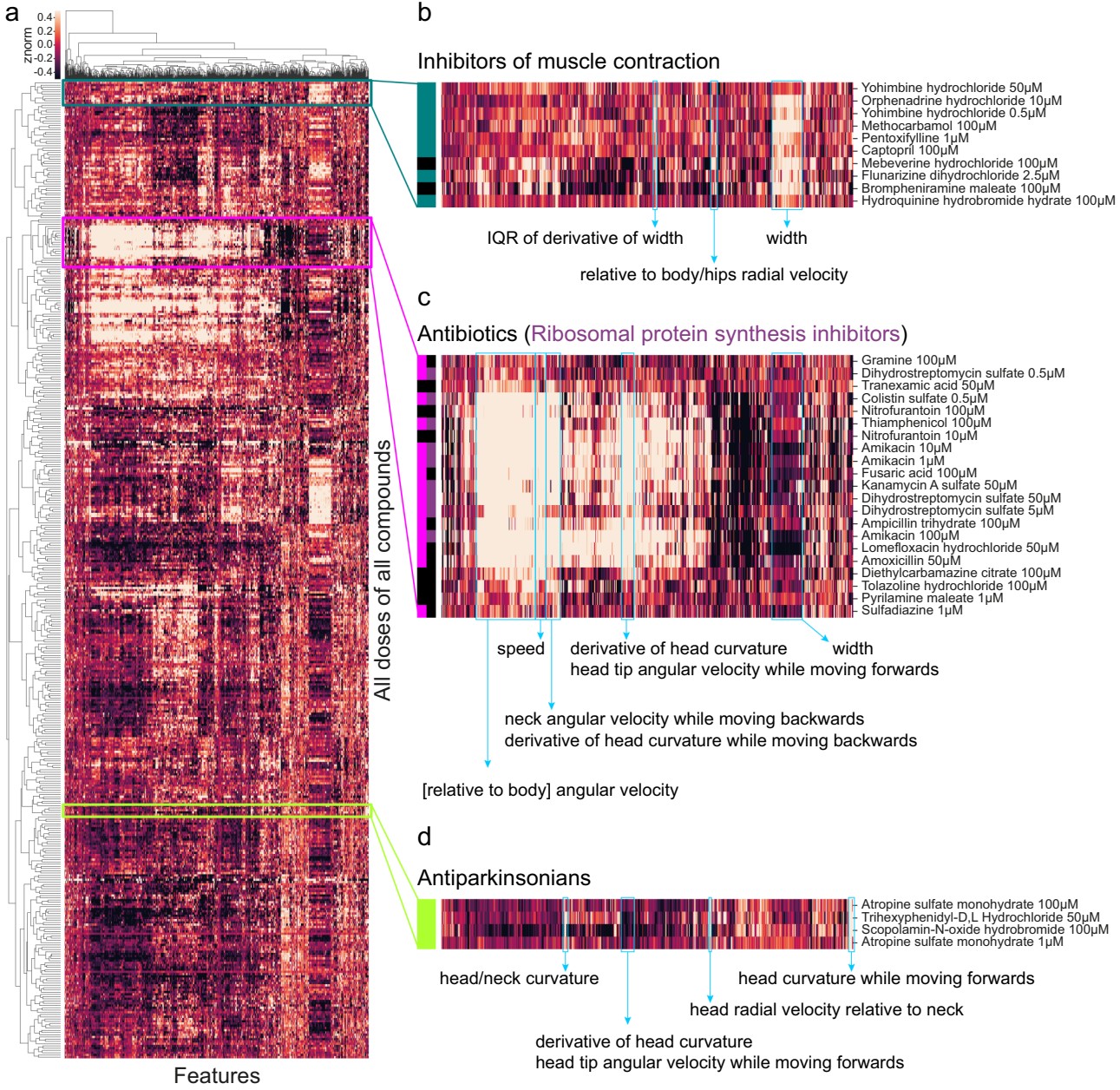

**Fig. 5 Worms respond differently to drugs with different modes of action. a** Heatmap of behavioural fingerprints of worms in response to treatment with drugs. Each row is the average phenotype of worms across multiple wells treated with the same compound at the same dose. Each column is a single behavioural feature. The colour indicates the z-normalised feature value. **b–d** Some compounds from the same class cluster together according to their behavioural response including inhibitors of muscle contraction (**b**), antibiotics (**c**), and antiparkinsonian (anticholinergic) drugs (**d**).

recordings and photostimulation was controlled using Loopbio's API for Motif software [https://github.com/loopbio/python-motifapi] in a script.

For the serial blue light stimulation experiments, the plates were continuously imaged for 43 min and 20 s in the following stimulation pattern: 5 min off, 20 x (10 s on at 100% intensity, 90 s off), 5 min off.

**Image processing and quality control**. Segmentation, tracking, and pose estimation over time was performed using Tierpsy Tracker. Each video was checked using Tierpsy Tracker's Viewer, and wells with visible contamination, agar damage, or excess liquid (from worm sorter, so that worms swim rather than crawl) were marked as bad and excluded from the analysis.

**Convolutional neural network to exclude nonworm objects**. We improved Tierpsy tracking by incorporating a CNN classifier after segmentation to exclude nonworm objects from being analysed and skewing the results.

In the video compression step at the beginning of the Tierpsy analysis pipeline, a segmentation algorithm detects putative worm objects according to a set of user-

defined parameters. The pixels in the frame that are further away than a threshold from any of the putative worms are set to 0, creating a "Masked Video". The objects selected by the masking algorithm are tracked throughout the video, but now if only they pass the filtering step powered by a CNN classifier.

The classifier was trained on a dataset of 43,561 grey-scale "masked" images measuring 80 × 80 pixels each, collected across several imaging systems in our lab. All images were manually annotated and objects were marked as either "worm" or "nonworm" by two independent researchers, so a consensus could be sought. The annotated dataset was split into training, validation, and test sets containing 80, 10, and 10% of the images, respectively, while keeping the classes balanced in each set. All images were pre-processed in two steps. First, the background pixels set to 0 by the masking algorithm were shifted to the top 95 percentile of the grey values in the unmasked area. This prevents the artificial edge between the masked and nonmasked area from disproportionately influencing the classifier. Second, all pixel values were scaled to the range of 0 to 1 by min-max normalisation, to reduce the influence of variable illumination and contrast in different imaging setups.

The architecture of the CNN is a shallower adaptation of VGG16[67], featuring eight convolution layers with 3 × 3 filter size and stride 1, each followed by a

rectified linear activation unit, four max-pool layers (filter size 2 × 2, stride 2) applied every two convolution layers, and a fully connected layer. Batch normalisation is applied to the third and seventh convolution layer to accelerate training by reducing internal covariate shift[68], and a Dropout layer is added before the fully connected layer to prevent overfitting[69]. In total, the CNN has about 1.78 million trainable parameters.

The CNN classifier was implemented in PyTorch 1.6, and was trained with the cross-entropy loss function and the Adam optimisation algorithm[70] at a learning rate of $10^{-4}$. It achieved an accuracy of 97.68% and F1 score of 97.98% as measured on the independent test set.

To improve performance at the inference step, we apply the CNN to a subset (one image per second) of all the images featuring the same putative worm object. This yields, per snapshot, the probability of the object to be a valid worm. If the median of this probability over time is higher than 0.5, the object is classified as a valid worm.

**Video processing with multiple wells**. Using multiwell plates for imaging significantly increased the experimental throughput, but also introduced challenges for data analysis as each video output contains 16 separate wells. Further software engineering was thus warranted to process multiwell videos, so that wells are detected and analysed separately.

To achieve this, we implemented an algorithm in Tierpsy Tracker that automatically detects multiple wells in a field of view and stores the coordinates of well boundaries. Briefly, we created a template that approximates the appearances of a well in the video, and replicated it on a lattice to simulate the grid of wells. The overall dimensions of the lattice are defined in Tierpsy's configuration file, but the lattice spacing parameters were chosen, via SciPy's differential evolution routine[71], to minimise the differences between the video's first frame (or its static background, if Tierpsy was instructed to calculate it) and the simulated grid of wells.

Automatic extraction of behavioural features was then performed on a per-worm basis, before worms were sorted into their respective wells based on their (x, y) coordinates in order to obtain well-averaged behavioural features.

**Data provenance**. Tracking multiwell plates complicates the handling of metadata, since there isn't a unique mapping between videos and experimental conditions. When well shuffling is performed using the liquid handling robot, the well contents in the imaging plate also needs to be tracked. To handle experimental metadata for imaging with the camera arrays, the records that need to be compiled manually during the experiments was standardised and an open-source module in Python (https://github.com/Tierpsy/tierpsy-tools-python/hydra) was developed to combine the experimental records to create a full metadata table with the experimental conditions for each well (Supplementary Fig. 3).

The experimental records are typically compiled in the form of csv files. In each tracking day, the experimenter needs to record: i) information about the media type and the bacterial food present on the imaging plates, and the worm strains that were dispensed into the wells of the plates (this is recorded in a summarized way in the *wormsorter.csv* file), ii) information about the experimental runs, including the unique IDs of the imaging plates, the instrument name where each plate was imaged, and the environmental conditions (*manual_metadata.csv*), iii) if applicable, information about the contents of the compound source plates (*sourceplate.csv*) and the mapping between imaging plates and source plates (if the liquid handling robot was used for column shuffling, this mapping will be recorded automatically in the *robotlog.csv*; if there was no shuffling, this will be recorded in *imaging2source.csv*).

Using the functions in the *hydra* module, firstly a plate metadata table is created to contain all the well-specific experimental conditions for every well of each unique imaging plate, including the compound contents if applicable. Then, the information about the experimental runs is merged with the plate metadata to create a final metadata table with the complete experimental conditions for every recording of every well. At this stage, the video filenames are also matched to the sample based on the camera array instrument ID. For example, scripts showing metadata handling, see https://github.com/Tierpsy/tierpsy-tools-python/tree/master/examples/hydra_metadata.

**Statistics and reproducibility**. Each well contains multiple worms (either 2 or 3, indicated in relevant captions) but worm identity is not necessarily maintained across the duration of tracking. We therefore use the number of wells as the sample size for statistical analysis rather than the number of worms. All experiments were repeated across at least three independent tracking days. Feature data and analysis scripts are available on Zenodo[72].

**Analysis of time-resolved response to photostimulation**. Tierpsy Tracker[17] was used to calculate a set of 3076 summary features for each well for each non-overlapping 10 s interval of the 6-minute stimulus recording (with three 10-second blue light pulses starting at 60, 160, and 260 sec). Samples where more than 40% of the features failed to be calculated were excluded from the analysis, and so was any feature that failed to be calculated for more than 20% of the samples in any of the 10 s intervals. Missing values were then imputed by averaging the valid values within each time interval. The feature matrix (all wells, in all time intervals) was then scaled by applying z-normalisation. Principal Components were then

**Table 1 Tested parameter grid for random forest classifier.**

| Parameter | Values | Selected |
|---|---|---|
| n_estimators | 200:200:2000 | 1000 |
| max_features | 'auto', 'sqrt' | 'sqrt' |
| max_depth | 10:10:110, None | 50 |
| min_samples_split | 2, 5, 10 | 2 |
| min_samples_leaf | 1, 2, 4 | 2 |

calculated using the whole feature matrix. Figure 3a shows a density plot of the measurements collected in the 10 s immediately before (left) and immediately after (right) a 10-second stimulus, projected onto the plane defined by the first two principal components.

To investigate the response to photostimulation with higher temporal resolution, Tierpsy Tracker[17] was used to detect the motion mode (forwards, backwards, stationary) of each worm over time. To calculate the fraction of worms in each motion mode over time (Fig. 3b), the number of worms in each motion mode at each time point in each well was divided by the total number of tracked worms at each time point in each well. This gave the fraction of worms in each motion mode, at each time point, for each well, so that an average could be taken across all wells. The 95% confidence interval for the average was obtained by nonparametric bootstrap ($n = 1000$ resamplings, with replacement).

For the longer experiments in Fig. 3c–e, the motion mode detected by Tierpsy Tracker for each worm overtime was first down-sampled to 0.5 Hz by dividing the video into nonoverlapping two seconds intervals and taking the prevalent motion mode in each interval. The fraction of the worm population in each motion mode over time was calculated by counting the number of worms in each motion mode and then dividing by the total number of worms detected at each time point. The 95% confidence interval was calculated via nonparametric bootstrap by the seaborn Python library.

**Classification of wild isolates**. For the classification of the divergent set we used a random forest classifier as implemented in scikit-learn[73]. For feature selection, we used recursive feature elimination with a random forest estimator (RFE), as implemented in scikit-learn[73]. We started by splitting the data randomly in a training/tuning set and a test set, with 20% of the data from each strain assigned to the test set. We used the training/tuning set for feature selection. We tried specific candidate feature set sizes $\{2^i, \text{ for } i = 7{:}11\}$. For each size, we performed cross-validation and: i) used each training fold to select N features and train a classifier with the selected features; ii) used each test fold to estimate the classification accuracy. We repeated the process 20 times to get statistical estimates of the mean cross-validation accuracy for each size and selected the best performing size $N_{best}$. We then selected $N_{best}$ features using the entire training/tuning set and used this set for downstream analysis. At a second stage, we tuned the hyperparameters of the random forest classifier using grid search with cross-validation as implemented in scikit-learn[73] with the grid shown in Table 1. The best-performing parameters are reported in Table 1. Finally, we trained a classifier on the entire training set using the selected features and hyperparameters and used it to make predictions on the test set.

**Reporting summary**. Further information on research design is available in the Nature Research Reporting Summary linked to this article.

## Data availability
The datasets (Tierpsy features, tracking data, and metadata) produced in this study are available on Zenodo: https://doi.org/10.5281/zenodo.5121521[72]. Any remaining information can be obtained from the corresponding author upon reasonable request.

## Code availability
General tools used for analysis and statistical tests are available on GitHub (github.com/Tierpsy/tierpsy-tools-python). Tracking software and source code is available at https://github.com/Tierpsy/tierpsy-tracker. The API for Motif is available at https://github.com/loopbio/python-motifapi and for the image store format at https://github.com/loopbio/imgstore.

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

## Acknowledgements

This project has received funding from the European Research Council (ERC) under the European Union's Horizon 2020 research and innovation programme (Grant agreement No. 714853) and was supported by the Medical Research Council through grant MC-A658-5TY30. This work was supported by a Research Grant from HFSP (Ref.-No: RGP0001/2019). AMR was supported by a BBSRC CASE studentship part-funded by Syngenta.

## Author contributions

Performed experiments: I.B., L.F., A.M.R., T.J.O., S.S.D. Analysed data: I.B., L.F., E.M., S.S.D. Supervised research: E.C.A., A.E.X.B. Built software: L.F., E.M., Z.L., M.H., J.R.S. Wrote the paper: I.B., L.F., M.H., J.R.S., E.C.A., S.S.D., A.E.X.B.

## Competing interests

The authors declare the following competing interests: MH and JRS are employees/owners of LoopBio. All other authors declare no competing interests.
