## [Transparent Peer Review File · Communications Biology]

Reviewers' comments:

Reviewer #1 (Remarks to the Author):

Barlow et al. have developed a robust multi-camera animal tracking system with improvements in throughput to the current state of the art. They demonstrate the systems utility with four different behavioral studies on *C. elegans*: behavioral variability across a small library of wild isolates, a stimulus-response experiment using blue light, phenotyping a number of *C. elegans* ALS models, and a pharmacological behavioral study. While the instrument is an integration and iteration of current technology, it is well demonstrated by a variety of experimental measurements and results. Each individual project reports novel results and overall the paper will be useful to a wide community interested in quantitative animal behavior.

I found the paper to be well written and technically sound. The materials and methods are complete. However I have a few comments.

One of the major improvements using this system is throughput. However, I don't see the number of worms, "N" measured, presented anywhere in the manuscript. Authors should include this for all data presented. In addition I would be interested in how sensitive some of the statistical testing is on the number of worms measured (but this isn't necessary for publication).

The authors adequately reference the *C. elegans* animal tracking literature. However, since they emphasize the lack of multi-camera systems used in the *C. elegans* field, I think it would be appropriate to mention that multi-camera systems are more common in the study of other organisms such as insects, fish, and mice.

There are a couple of "black boxes" or proprietary elements of the system coming from Loopbio. The capability and how they are used are well described, but given this is a technical paper, it would be appreciated if the MOTIF solution is necessary and alternatives difficult or if alternatives are readily available.

Minor:

1) Abstract: The authors write "Current imaging systems are limited either in spatial resolution or throughput." In what sense? I don't find this to be a fair statement. Plenty of current systems have been used to make significant discoveries in "neuroscience, genetics, disease modelling, and drug discovery,"

2) L. 151. "LoopbBio" should be "LoopBio"

2) L. 285. "...resulting in a total of 9228 features for each well." I understand that image capturing is on the basis of area (e.g. per well), but wouldn't it be more natural to report measurements "per worm?"

3) L.432. "Tierpsy256" typo?

4) Fig 5. I suppose not much can be done and it may be just how my PDF copy was rendered, but I found it very difficult to parse Fig 5.

Reviewer #2 (Remarks to the Author):

Phenotypic screens in *C. elegans* have a strong potential for drug discovery, yet in the field of

neuroscience, and in particular when the phenotypes of interests are behaviors, a major bottleneck relates to the quantification of these behaviors. Recent developments in tracking systems, including major contributions by Brown and colleagues, have considerably increased the efficiency and quality of automated tracking, feature extraction and behavioral characterization. However, some bottlenecks still hindered an efficient implementation of the model in a drug screening context. These bottlenecks include the throughput of data acquisition in multiple conditions and the compatibility with standard multi-well format for direct coupling with existing drug libraries, both of which are directly tackled in the present article by Barlow and colleagues. They describe a multiplexed machine-vision-based tracking system enabling (i) the high-content analysis of *C. elegans* behavior in several 96-well plates at a time and (ii) the implementation of temporally controlled blue light stimuli. The authors further report the validation of this system by showing its utility for comparisons across multiple genotypes and upon drug treatment. The system is really impressive. The design is well justified and the validation convincing. I have only a few concerns/suggestions, which I hope the authors will be able to address.

Main point

1) I have only one main concern, which relates to the way the present manuscript describes the paraquat treatment in a previously published article [ref 43] and the parallel made between blue-light escape (here) and the paraquat effect [ref 43]. In ref 43, they described the results of an assay in which paraquat-treated worms 'disappear' from the plate after an overnight period (presumably going on the petri dish walls) and interpret this observation as a 'paraquat escape'. But in the same paper, they also demonstrate that a paraquat-induced oxidative stress damages neurons and that the same overnight treatment causes a major defect in nose touch assays. Most likely, the 'paraquat escape' effect is due to oxidative damages in the sensory-behavior circuit, which normally prevents worm from climbing on the dish wall (which may presumably include mechanosensation, chemosensation and hygro-sensation). Maybe paraquat impacts sensory neurons or sets the worms in a fast moving mode such that animals are more likely to go on the wall 'by accident'. Can we really call this "an escape response from noxious paraquat" (line 414-415 in the present ms)? This phrasing suggests an ability to sense paraquat as noxious and escape it via a taxis-like behavior. In contrast, the blue light response here is clearly a direct sensory-behavior response. And the assay takes place on a dramatically different timescale. I really think that the authors, should describe it differently. Furthermore, they don't need to look for a parallel and try to "reproduce" (line 416) this previous study. Their high-content behavioral characterization approach (including stimulus-evoked response) is by-design prone to detect subtle defects as it encompasses many parameters.

Minor points

- 2) Lines 99-101. It is essential to have some more explanations about the geometry of the IR illumination system (e.g. LED density, light angles) and what optical trick is used to homogenize the background light (I am assuming it is needed because LED are very punctual light sources and even 65 mm away, I am pretty sure light will be very inhomogeneous without a well-thought design).
- 3) Same thing for the blue illumination: the geometry (including LED orientation) is not very clear from Fig 1 only. This is essential to evaluate the nature of the blue light stimulations.
- 4) How homogenous is the blue light received across wells? It is essential to report the actual (measured) light intensity at the worm level, so readers can relate the behavior to previously published work and assess the ability to adapt the system for optogenetics. Moreover, can the light intensity be tuned by the user?
- 5) Can the system be adapted to 24 or 48-well plates (this is a point of curiosity, not a request)? There might be applications where the handling of worms upstream is more challenging and represents a stronger limitation than the number of drugs to be tested; or applications where the drug is expensive and more observation per medium volume unit is preferable. In these cases, it might be suitable to record more worms per well (notably with a higher % of area available for recording) and less wells.
- 6) Line 280: It would be nice to already know here the number of worms per well without referring to the method section.

- 7) With only 3 worms per well and 41% area, what fraction of the time one will record 3, 2, 1 and 0 worm respectively?
- 8) It is contradictory to justify the choice of the 245 drugs based on their accumulation in the worm (line 429) and later argue that some drugs have no effect because there is no uptake (line 461-462).
- 9) All the observation numbers should be more explicitly mentioned in Figure and/or legends.
- 10) Line 349: "a single dose of photostimulation" is really unclear for several reasons, but the main issue is that the word "dose" could imply that you used different intensities in different stimuli trains (which I think is not the case). What about "the first photostimulus" or "a single photostimulus" or something similar?
- 11) Line 484: reversal and accelerations are very different behaviors engaging different neurons: why would you have strong expectations that the plasticity is the same?
- 12) Line 536: please also specify the actual starting and target weight of the media, it will be more useful.

Reviewer comments in blue

Responses in black

In addition to the points below, we have updated Figure 3B: we noticed a bug in our code that caused worms treated with chemical compounds to be erroneously combined with non-treated worms and shown in the figure. Treated worms have now been removed. Figure 3B-E has been updated as we collected more data using the same protocol. Our findings do not change.

Figure 5 has also been modified, as we elected to drop from the dataset a small number of drugs that, unlike the rest of the library, were dissolved in water rather than in DMSO. We have updated the text to reflect the new Figure 5.

Reviewers' comments:

Reviewer #1 (Remarks to the Author):

Barlow et al. have developed a robust multi-camera animal tracking system with improvements in throughput to the current state of the art. They demonstrate the systems utility with four different behavioral studies on *C. elegans*: behavioral variability across a small library of wild isolates, a stimulus-response experiment using blue light, phenotyping a number of *C. elegans* ALS models, and a pharmacological behavioral study. While the instrument is an integration and iteration of current technology, it is well demonstrated by a variety of experimental measurements and results. Each individual project reports novel results and overall the paper will be useful to a wide community interested in quantitative animal behavior.

I found the paper to be well written and technically sound. The materials and methods are complete. However I have a few comments.

One of the major improvements using this system is throughput. However, I don't see the number of worms, "N" measured, presented anywhere in the manuscript. Authors should include this for all data presented. In addition I would be interested in how sensitive some of the statistical testing is on the number of worms measured (but this isn't necessary for publication).

For each experiment, we now include the number of worms, wells, and tracking days for each experiment in the captions and as a table in the supplementary material section. We have also investigated how the number of significant features increases with the number of wells measured (Supplementary figure 6).

The authors adequately reference the *C. elegans* animal tracking literature. However, since they emphasize the lack of multi-camera systems used in the *C. elegans* field, I think it would be appropriate to mention that multi-camera systems are more common in the study of other organisms such as insects, fish, and mice.

We now mention other multicamera imaging systems in the discussion and compare Motif to alternative methods of recording from multiple cameras.

There are a couple of “black boxes” or proprietary elements of the system coming from Loopbio. The capability and how they are used are well described, but given this is a technical paper, it would be appreciated if the MOTIF solution is necessary and alternatives difficult or if alternatives are readily available.

There are other methods available for recording data from multiple cameras and we have now included reference to these in the discussion. We have also provided more details on the Loopbio elements.

Minor:

1) Abstract: The authors write “Current imaging systems are limited either in spatial resolution or throughput.” In what sense? I don’t find this to be a fair statement. Plenty of current systems have been used to make significant discoveries in “neuroscience, genetics, disease modelling, and drug discovery,”

We have rephrased this to say that increased throughput and resolution will enable new kinds of experiments.

2) L. 151. “LoopbBio” should be “LoopBio”

Thanks, this has been corrected.

2) L. 285. “...resulting in a total of 9228 features for each well.” I understand that image capturing is on the basis of area (e.g. per well), but wouldn’t it be more natural to report measurements “per worm?”

Because we cannot maintain worm identity through the course of tracking, we initially calculate the features at the level of ‘tracks’ (with the number of unique tracks typically much greater than the number of worms). Given the uncertainty in how these tracks map to worms, we only use a single averaged feature vector per well. We have now made this more explicit in the results section.

3) L.432. “Tierpsy256” typo?

We use Tierpsy256 as shorthand for the subset of 256 features that we selected in a previous paper (Javer et al. 2018, Phil Trans B). We have clarified this in the results section.

4) Fig 5. I suppose not much can be done and it may be just how my PDF copy was rendered, but I found it very difficult to parse Fig 5.

We have made the figure somewhat taller and increased the font size which will hopefully help with legibility.

Reviewer #2 (Remarks to the Author):

Phenotypic screens in *C. elegans* have a strong potential for drug discovery, yet in the field of neuroscience, and in particular when the phenotypes of interests are behaviors, a major bottleneck relates to the quantification of these behaviors. Recent developments in tracking

systems, including major contributions by Brown and colleagues, have considerably increased the efficiency and quality of automated tracking, feature extraction and behavioral characterization. However, some bottlenecks still hindered an efficient implementation of the model in a drug screening context. These bottlenecks include the throughput of data acquisition in multiple conditions and the compatibility with standard multi-well format for direct coupling with existing drug libraries, both of which are directly tackled in the present article by Barlow and colleagues. They describe a multiplexed machine-vision-based tracking system enabling (i) the high-content analysis of *C. elegans* behavior in several 96-well plates at a time and (ii) the implementation of temporally controlled blue light stimuli. The authors further report the validation of this system by showing its utility for comparisons across multiple genotypes and upon drug treatment. The system is really impressive. The design is well justified and the validation convincing. I have only a few concerns/suggestions, which I hope the authors will be able to address.

Main point

1) I have only one main concern, which relates to the way the present manuscript describes the paraquat treatment in a previously published article [ref 43] and the parallel made between blue-light escape (here) and the paraquat effect [ref 43]. In ref 43, they described the results of an assay in which paraquat-treated worms ‘disappear’ from the plate after an overnight period (presumably going on the petri dish walls) and interpret this observation as a ‘paraquat escape’. But in the same paper, they also demonstrate that a paraquat-induced oxidative stress damages neurons and that the same overnight treatment causes a major defect in nose touch assays. Most likely, the ‘paraquat escape’ effect is due to oxidative damages in the sensory-behavior circuit, which normally prevents worm from climbing on the dish wall (which may presumably include mechanosensation, chemosensation and hygrosensation). Maybe paraquat impacts sensory neurons or sets the worms in a fast moving mode such that animals are more likely to go on the wall ‘by accident’. Can we really call this “an escape response from noxious paraquat” (line 414-415 in the present ms)? This phrasing suggests an ability to sense paraquat as noxious and escape it via a taxis-like behavior. In contrast, the blue light response here is clearly a direct sensory-behavior response. And the assay takes place on a dramatically different timescale. I really think that the authors, should describe it differently. Furthermore, they don’t need to look for a parallel and try to “reproduce” (line 416) this previous study. Their high-content behavioral characterization approach (including stimulus-evoked response) is by-design prone to detect subtle defects as it encompasses many parameters.

This is a good point. We have clarified the results and discussion as suggested. We no longer draw a direct parallel between the earlier results and ours and emphasise instead that in both cases no differences are observed at baseline, but that aversive conditions, perhaps operating through very different mechanisms, can be useful revealing otherwise cryptic phenotypes.

Minor points

2) Lines 99-101. It is essential to have some more explanations about the geometry of the IR illumination system (e.g. LED density, light angles) and what optical trick is used to homogenize the background light (I am assuming it is needed because LED are very punctual light sources and even 65 mm away, I am pretty sure light will be very inhomogeneous without a well-thought design).

The light panel we use is an edge-lit LED panel provided by LoopBio and yields fairly uniform illumination. In the edge-lit configuration, individual LEDs are attached to the frame of the panel, and shine horizontally into a light guide plate. The light guide plate, together with a diffuser, homogenizes the light minimising the risk of bright spots. To improve light collimation, we use two computer monitor privacy filters placed at right angles. We have added details about the design and the uniformity of the bright field illumination to the methods section.

- 3) Same thing for the blue illumination: the geometry (including LED orientation) is not very clear from Fig 1 only. This is essential to evaluate the nature of the blue light stimulations.
- 4) How homogenous is the blue light received across wells? It is essential to report the actual (measured) light intensity at the worm level, so readers can relate the behavior to previously published work and assess the ability to adapt the system for optogenetics. Moreover, can the light intensity be tuned by the user?

We have included a more detailed schematic showing the LED orientation as well as a quantification of the blue light intensity distribution in the supplementary information. The light intensity can be tuned by the user on a unit-by-unit (i.e. plate-by-plate) basis thanks to a control slider in the Motif software interface. The intensity of the photostimuli can also be selected via the API when running complex photostimulation experiments. We have added this point to the methods.

- 5) Can the system be adapted to 24 or 48-well plates (this is a point of curiosity, not a request)? There might be applications where the handling of worms upstream is more challenging and represents a stronger limitation than the number of drugs to be tested; or applications where the drug is expensive and more observation per medium volume unit is preferable. In these cases, it might be suitable to record more worms per well (notably with a higher % of area available for recording) and less wells.

Thank you for bringing up this point. We designed the system with other well formats in mind. Since the field of view covers the entire well plate, any number of wells can be used. However, for some configurations, wells may be split across two cameras (e.g. with a 12 well plate). The videos could be stitched in principle, but that is not currently implemented so for most purposes we would stick to 24, 48, and 96 well plates. We have included some sample images using other numbers of wells in the supplement and mention this fact in the results. Tierpsy is in the process of being configured to handle other well numbers in terms of well identification and metadata handling.

- 6) Line 280: It would be nice to already know here the number of worms per well without referring to the method section.

We have added worm, well, and plate numbers for all experiments in the captions, and in tables in the supplementary materials.

- 7) With only 3 worms per well and 41% area, what fraction of the time one will record 3, 2, 1 and 0 worm respectively?

The worms can indeed hide in the shaded region at the edge of each well. The number of worms simultaneously visible in each well can vary by strain and treatment. For example, when only 2 worms are dispensed per well, 2, 1, and 0 N2 worms will be recorded

approximately 40%, 40%, and 15% of the time. These figures are considerably different for CB4856, where 2, 1, 0 worms are recorded for approximately 9%, 36%, and 53% of the time. Increasing the number of worms per well to 3 improves these figures: we recorded 3, 2, 1, and 0 N2 worms for respectively 39%, 41%, 15%, 3% of the time. We observed similar results for the ALS strains. We have now added a figure showing what fraction of the time n worms were recorded, and the distribution of the maximum number of worms ever recorded simultaneously in a well (Supplementary Figure S5). We now address this explicitly in the main text.

8) It is contradictory to justify the choice of the 245 drugs based on their accumulation in the worm (line 429) and later argue that some drugs have no effect because there is no uptake (line 461-462).

We agree, and following a more careful reading of the information supplied by the vendor we have removed the statement that the drugs were chosen based on their accumulation.

9) All the observation numbers should be more explicitly mentioned in Figure and/or legends.

We have added the number of worms, wells, and plates/days to the figure captions for each experiment.

10) Line 349: “a single dose of photostimulation” is really unclear for several reasons, but the main issue is that the word “dose” could imply that you used different intensities in different stimuli trains (which I think is not the case). What about “the first photostimulus” or “a single photostimulus” or something similar?

We agree that the phrasing we used is unclear, and modified it to read “a single photostimulus”.

11) Line 484: reversal and accelerations are very different behaviors engaging different neurons: why would you have strong expectations that the plasticity is the same?

We agree and have removed the statement about expectations.

12) Line 536: please also specify the actual starting and target weight of the media, it will be more useful.

We have added the starting and target weights.

REVIEWERS' COMMENTS:

Reviewer #1 (Remarks to the Author):

I'm happy with the responses to the comments and changes made to the manuscript. Onwards.

Reviewer #2 (Remarks to the Author):

The authors have addressed all my concerns in a very carefully revised manuscript. This is really an impressive methodology and I fully support the publication of this great article.